# Bioaccessibility and Cellular Uptake of Carotenoids Extracted from *Bactris gasipaes* Fruit: Differences between Conventional and Ionic Liquid-Mediated Extraction

**DOI:** 10.3390/molecules26133989

**Published:** 2021-06-30

**Authors:** Leonardo M. de Souza Mesquita, Daniella Carisa Murador, Bruna Vitória Neves, Anna Rafaela Cavalcante Braga, Luciana Pellegrini Pisani, Veridiana Vera de Rosso

**Affiliations:** 1Department of Biosciences, Federal University of São Paulo (UNIFESP), Silva Jardim Street 136, Vila Mathias, Santos, SP 11015-020, Brazil; mesquitalms@gmail.com (L.M.d.S.M.); danicarisa@hotmail.com (D.C.M.); bruna.vitoria@unifesp.br (B.V.N.); anna.braga@unifesp.br (A.R.C.B.); lucianapisani@hotmail.com (L.P.P.); 2Department of Exact and Earth Sciences, Campus Diadema, Federal University of São Paulo (UNIFESP), Diadema, SP 09972-270, Brazil; 3Nutrition and Food Service Research Center, Federal University of São Paulo (UNIFESP), Silva Jardim Street 136, Santos, SP 11015-020, Brazil

**Keywords:** [C_4_mim][BF_4_], Caco-2 cells, Amazonian fruit, pigments, all-*trans*-β-carotene, all-*trans*-lycopene

## Abstract

Currently, on an industrial scale, synthetic colorants are used in many fields, as well as those extracted with conventional organic solvents (COSs), leading to several environmental issues. Therefore, we developed a sustainable extraction and purification method mediated by ionic liquids (IL), which is considered an alternative high-performance replacement for COSs. Carotenoids are natural pigments with low bioaccessibility (BCT) and bioavailability (BV) but with huge importance to health. To investigate if the BCT and cellular uptake of the carotenoids are modified by the extraction method, we conducted a comparison assay between both extraction procedures (IL vs. COS). For this, we used the Amazonian fruit *Bactris gasipaes*, a rich source of pro-vitamin A carotenoids, to obtain the extract, which was emulsified and subjected to an in vitro digestion model followed by the Caco-2 cell absorption assay. The bioaccessibility of carotenoids using IL was better than those using COS (33.25%, and 26.84%, respectively). The cellular uptake of the carotenoids extracted with IL was 1.4-fold higher than those extracted using COS. Thus, IL may be a feasible alternative as extraction solvent in the food industry, replacing COS, since, in this study, no IL was present in the final extract.

## 1. Introduction

Carotenoids are pigments that are naturally present in many biomasses (fruits, algae, bacteria, and some animals), are important for human diets since they display several biological activities, and are considered an excellent food additive to enhance the nutritional benefits of foods [1]. These pigments are tetraterpenoids linked by conjugated double bonds, conferring high molecular hydrophobicity, thus justifying the use of conventional organic solvents (COSs) for their synthesis and extraction [2]. Choosing the type of solvent used for the extraction of a natural pigment is an important step to obtain the highest yields [3]. The yellow-reddish synthetic pigments, which mimic the color of natural carotenoids, are mainly produced using petroleum derivative solvents, which are considered hazardous to the environment and human health [4]. Besides, the steps involved in the industrial extraction mediated by COS have huge environmental effects, such as the tremendous energy consumption, long time period of homogenization between the natural source and solvents, and also the large number of solvents needed to recover high yields of the compounds [5]. On the other hand, alternative solvents, such as ionic liquids (ILs), could be a safer, eco-friendly, and feasible alternative for obtaining natural pigments [6]. Simply, ILs are designer solvents formed by salts with a low charge density and low symmetry among their ions [7]. Additionally, ILs are considered high-performance solvents since they usually promote high extraction yields when compared to those of COS-mediated processes. This is an important characteristic, since one of the main drawbacks of the industrial sector is obtaining large yields of stable natural pigments [8,9]. 

In the current literature, some studies have already used IL to obtain natural carotenoids, reporting excellent results regarding the biocompatibility, yield, and thermal stability [10,11,12]. Additionally, from an environmental perspective, ILs have emerged as promising solvents to replace COS since they can be recycled in new processes [6], reducing the environmental impact caused by the disposal of underexploited raw materials [13]. Furthermore, the protocols used to produce most usual synthetic food colorants are unsafe, which promotes the confidence of new consumers [14,15]. Additionally, synthetic dyes are produced by photocatalytic reactions, without recycling of the raw materials (petroleum derivative chemicals), which goes against sustainable chemistry principles, as exposed by the European economic plan based on the circular economy for a cleaner and more competitive society [16].

Considering the health benefits of food pigments, the bioaccessibility (BCT) and bioavailability (BV) of lipophilic bioactive molecules can be affected by several endogenous (genetics and sex or age of the consumer) and exogenous factors (food processing, soluble fiber, carotenoid deposition, thermal processing, and amount and type of fat) [17]. Unfortunately, carotenoids have a low absorption rate in the body, so some strategies have been developed to aid in their absorption, such as the use of new excipient foods, nanoencapsulation, and emulsions [18,19]. To be bioaccessible, carotenoids must be released from their food matrix in the stomach and incorporated into micelles containing pancreatic lipases and bile salts [20]. Nevertheless, depending on the food matrix, this phenomenon can be impaired since the initial physical state of the carotenoid in the biomass is an often-overlooked factor that could influence the release and subsequent carotenoids’ BV [21]. Therefore, the addition of natural carotenoids as food supplements may be an interesting strategy for increasing their absorption since, under this condition, the carotenoids would already be free from the food matrix, enhancing their BCT and BV [22].

Additionally, carotenoids represent an important source of liposoluble vitamin A (VA). More specifically, the *trans* isomers of α-carotene, β-carotene, and β-cryptoxanthin can be converted into VA in the human body [23]. The deficiency of this vitamin is a huge problem in many worldwide populations and is associated with severe health problems, namely, diarrhea, dermatitis, and stress [23]. Additionally, some recent studies have reported that VA is involved in body weight maintenance, as it assists in controlling energy homeostasis by modulating the production of leptin and inflammatory cytokines [24]. Taking this into account, we chose to investigate the BCT and Caco-2 cellular uptake of carotenoids present in the Amazonian *Bactris gasipaes* fruits (peach palm — Arecaceae family) since it is considered a pro-VA carotenoid-rich fruit [25], and is usually wasted from agribusiness since the palm-heart present in this plant is the main target for the national and international market, with higher added value compared to its fruits [10]. Thus, the non-sustainable exploitation of *Bactris gasipaes* fruits contributes to the loss of genetic biodiversity in the Amazon region and underexploitation of its economic potential, a severe consequence considering the current global situation.

Therefore, despite the unquestionable success of IL in the extraction of natural compounds [6,26], there are some knowledge gaps regarding the evaluation of the BCT and cellular uptake from natural compounds obtained with IL-mediated extractions [27]. In our recent previous work [10], a complete and optimized process mediated by IL was developed to recover high yields of carotenoids from B. *gasipaes* (Figure 1). It was proved that the sustainability of the IL-mediated extraction is at least 2-fold better than the COS process (using acetone and ether mixtures as extraction solvents), proved by the low carbon footprint and by the application of the circular economy principle, as recommended by the most prestigious journals in the field. Additionally, besides the selectivity that IL promotes in the extraction process, the carotenoids extracted with IL are more thermally stable and have high antioxidant activity. Furthermore, the polishing (separation of IL from the carotenoids’ extract) was also optimized, with 94% recovery of IL used (Appendix A). Thus, to discuss the consequence of the BCT and cellular uptake from carotenoids extracted with IL, in this work, we evaluated whether there is a difference in BCT and Caco-2 cellular uptake of carotenoids extracted with IL and COS. Additionally, the knowledge about the bioactive compounds of peach palm fruits could help implement public policies that can support sustainable agriculture/forestry incentives.

## 2. Materials and Methods

### 2.1. Raw Materials

*Bactris gasipaes* fruits were acquired from a sustainable agriculture cooperative located in northeastern Brazil (Ilhéus-Bahia-Brazil — 14°50′00.47″ S, 39°01′51.98″ W). Subsequently, the fruits were sanitized in tap water with the removal of the seeds. The fruit pulp was immediately frozen at −40 °C, lyophilized for 48 h, and then stored at −40 °C to preserve the carotenoid content for further analysis.

### 2.2. Chemicals

The ILs 1-butyl-3-methylimidazolium tetrafluoroborate ([C_4_mim][BF_4_]) and polysorbate 80 (Tween 80) were purchased from Sigma-Aldrich^®^, with >99% purity. Methanol (100%), petroleum ether (100%), diethyl ether (100%), and acetone (100%) were purchased from Labsynth^®^, and methyl tert-butyl ether (MTBE) was purchased from Honeywell^®^ (10% — LC grade). Sunflower oil was purchased from a commercial market in Brazil. Standards of all-*trans*-β-carotene (Sigma^®^ 1065480) and all-*trans*-lycopene (Sigma^®^ 1370860) were used for the quantification assays. For cultivation and treatment of Caco-2 cells, fetal bovine serum (Sigma^®^ F7524), Dulbecco’s modified Eagle’s medium-high glucose (Sigma^®^ D5796), MEM nonessential amino acid solution (Sigma^®^ M7145), and penicillin-streptomycin solution (Sigma^®^ P4333) were used.

### 2.3. Solid-Liquid Extraction

#### 2.3.1. Conventional Extraction—Acetone

The conventional solvent extraction of carotenoids was performed according to de Rosso and Mercadante (2007) [25]. Briefly, 1 g of the freeze-dried fruit samples’ pulp was homogenized with acetone (100%), and the carotenoids extracted were transferred to petroleum ether: diethyl ether (2:1 *v*:*v*) solution (liquid-liquid extraction). The upper phase, consisting of ether and carotenoids, was evaporated to dryness using a rotary evaporator at temperatures below 37 °C under vacuum (15 min). The extracted carotenoids were stored at −40 °C for further analysis. The results obtained were performed in triplicate, and the standard deviations were calculated.

#### 2.3.2. Alternative Extraction—Ionic Liquids

The carotenoids extracted with an IL were obtained according to de Souza Mesquita et al. (2019) [10]. Briefly, the procedure with the IL [C_4_mim][BF_4_]-ethanolic solution was optimized to achieve the maximum content of carotenoids from the peach palm freeze-dried pulp. The samples were submitted to ultrasonic-assisted extraction for 12 min with a solid-liquid ratio and ethanol-IL ratio of 1:1 (*w*:*w*). Carotenoid polishing was performed by thermal precipitation of the [C_4_mim][BF_4_] at −80 °C with complete separation of the IL and recovery of the solvents, and the recovery of IL was calculated at 94%. The carotenoid extract was lyophilized and stored at −40 °C for further analysis. This extract is considered more eco-friendly compared to the acetonic extract since is based on the green chemistry principles, circular economy, biorefinery, and lower environmental impact expressed by the low carbon footprint.

#### 2.3.3. Quantification of [C_4_mim][BF_4_] in Carotenoid Extract

The residual of IL present in the carotenoids’ extract was quantified by UV spectroscopy (Carry 5.0, Varian) using a quartz cuvette (d = 1 cm) at λ_max_ 211.1 nm (maximum absorption wavelength of the IL). The IL was quantified using [C_4_mim][BF_4_] as an external standard for quantification with a calibration curve of seven points (0.315 — 10 µg/mL — r^2^ = 0.9992), as exposed in Cao et al. (2014) [28]. To evaluate the accuracy of the quantification, the extracts that were submitted to the polishing step (i.e., those used in the simulated digestion process), and extracts without the polishing step (control) were evaluated. For this, an initial IL solution of 500 µg/mL was used to perform the extractions, after proper polishing steps (or without polishing for control extracts), and since the [C_4_mim][BF_4_] is not soluble in ether, a liquid-liquid extraction using water: petroleum ether (3:2 *v*:*v*) was executed. For both extracts, to separate carotenoids from the IL, the aqueous phase (composed by ethanol and IL) was recovered, concentrated in a rotary evaporator (to eliminate the ethanol), and then quantified by UV in aqueous solution. All the analysis was performed in triplicate, and the results were expressed in mean ± standard deviation of the mean (%).

### 2.4. Bioaccessibility (BCT) and Cellular Uptake Assay

#### 2.4.1. Sample Preparation: Emulsified Carotenoid Extracts

Considering the solubility of the carotenoids in oily media, both extracts (mediated by acetone and [C_4_mim][BF_4_]) were firstly resuspended in sunflower oil with the aid of magnetic stirring (37 °C), following by filtration (Whatman N°.1 filter paper), and the resulting oil (with known carotenoids’ concentration) was used to prepare the emulsion. No carotenoid crystal was precipitated in the oil since a translucid solution was achieved. Appendix A highlights the whole method performed in this work to assess the BCT and cellular uptake of carotenoids from both extracts, which were performed in triplicate. The emulsions were carried out to simulate a pigmented food, which was prepared according to Salvia-Trujillo et al. (2017), with some modifications [19]. Briefly, the samples were composed of 18% (*w*:*w*) sunflower oil, 2% (*w*:*w*) Polysorbate 80 (Tween 80) as a surfactant, and 80% (*w*:*w*) distilled water. The emulsions were prepared using ultra-disperser homogenization (16,000 rpm/4 min).

#### 2.4.2. Microstructure of the Emulsion’s Droplets

As exposed by Salvia-Trujillo et al. (2017) [19], the microstructure of the initial emulsions (before bioaccessibility assay), as well as after the oral, gastric, and intestinal stages, were evaluated by light microscopy (Zeiss Primo Star, Oberkochen, Germany), for both IL and acetone extracts. Subsequently, these images were scanned by an image analysis system (AxionVision—Zeiss), which was utilized for the evaluation of the diameter size of the droplets. A total of 5 images were taken by the group in all stages, and 25 droplets were randomly measured in each image. The results were expressed as mean ± standard deviation of the mean (µm).

#### 2.4.3. In Vitro Digestion and Caco-2 Cellular Uptake

The emulsions were first submitted to a simulated in vitro digestion model according to Failla and Chitchumronchokchai (2005) [29]. Briefly, 5 g emulsions were homogenized with 10 mL of a basal salt solution (NaCl, 120 mol/L; CaCl_2_, 6 mmol/L; KCl, 5 mmol/L). The oral digestion phase was initiated with 6 mL of a solution of artificial saliva containing 106 u/mL α-amylase (Sigma^®^ A3176), followed by incubation at 37 °C for 10 min in an orbital shaker (150 rpm). Before starting the gastric digestion phase, the pH was adjusted to 2.5 with 1 M HCl followed by the addition of 2 mL pepsin (Sigma^®^ P7000; 50,000 units/mL in 100 mM HCl). The total volume was adjusted to 40 mL, and the solution was incubated for 1 h at 37 °C and 150 rpm. After this step, the pH was adjusted to 6.0 with 1 M NaHCO3, and the intestinal digestion phase was initiated with a porcine and ovine bile solution (3 mL; Sigma^®^ B8381; 40 mg/mL in 100 mM NaHCO_3_), 4000 u/mL porcine pancreatins (Sigma^®^ P1750), and 1000 u/mL lipases from porcine pancreas (Sigma^®^ L3126). Incubation was conducted for 2 h at 37 °C, and the pH was adjusted to 6.5 in 50 mL. After the completed in vitro digestion, the solution was centrifuged at 8,000 g for 60 min at 4 °C. The bioaccessible carotenoids were incorporated into micelles and were present in the supernatant after filtration of the centrifuged solution using a 0.22 µm filter. The ratio between micellar carotenoids and the determinant in the microemulsion before digestion refers to the carotenoid bioaccessibility. To evaluate the capitation of the micellar carotenoids in an in vitro model, Caucasian colon adenocarcinoma cells (Caco-2) were used. For the cell culture, we used Caco-2 cells purchased from the *Banco de Células do Rio de Janeiro*—BCRJ, which were treated for 15 days with Dulbecco’s modified Eagle’s medium-high glucose (Sigma^®^ D5796, DMEM, 81%), fetal bovine serum (Sigma^®^ F7524, SFB, 15%), Eagle′s minimum essential medium (MEM), non-essential amino acids solution (Sigma^®^ M7145, MEM, 2%), and penicillin-streptomycin solution (Sigma^®^ P4333 2%) in a flask (Sarsted^®^) to 25 cm^2^. The experiment was performed when the cells were in passage 21. The cells were counted by the Neubauer chamber, and 4 × 105 cells were placed in each flask. After they achieved 80% confluence, between 10 and 15 days, a medium containing 25% of the micellar fraction was applied to the Caco-2 cells, which were incubated under a controlled atmosphere of air/CO2 (95:5) at 37 °C for 4 h. Then, the flask was washed with 1 mL of phosphate-buffered saline-PBS (Gibco^®^), and 5 mL of pure DMEM were added and submitted to incubation for more than 6 h. After this period, the cells were collected and centrifuged at 8000× *g* for 5 min. The debris was freeze-dried, and the extraction was subsequently performed. The protein content of the cells was analyzed according to Bradford’s methodology [30]. The results were expressed as the uptake of carotenoids by Caco-2 cells (ng of carotenoids/mg cell protein).

### 2.5. Analysis of the Carotenoids by HPLC-PDA

The carotenoids from the micellar phase and Caco-2 cells were extracted with petroleum ether using an ultrasonic probe (400 W, 5 min), and the procedure was repeated until the samples became colorless. Then, an aliquot of the carotenoid extract was evaporated under N_2_ flow, dissolved in methanol:MTBE (1:2), and injected into a chromatographic system. The HPLC-PDA analysis was performed in a Shimadzu^®^ HPLC (Kyoto, Japan) equipped with quaternary pumps (LC-20AD), a degasser unit (DGU-20A5), a Rheodyne injection valve with a 20 µL loop, and a diode array detector (DAD) (SPD-M20A). HPLC analysis was performed according to de Souza Mesquita et al. (2020a). Briefly, the chromatographic profile was obtained using a C_30_ YMC column (3 µm, 250 × 4.6 mm i.d., Waters, Wilmington, MA, USA), with a mobile phase composed of a linear gradient of methanol:MTBE from 75:25 to 50:50 up to 25 min, then 50:50 until 65 min. The flow rate was fixed at 0.9 mL/min, and the controlled column temperature at 22 °C. The data profiles were processed at 450 nm [31]. The carotenoids were quantified using all-*trans*-β-carotene and all-*trans*-lycopene as external standards for quantification (r^2^ = 0.999).

### 2.6. Data Presentation and Statistical Analysis

The results are expressed as the mean ± standard deviation of the mean. All the assays were performed in triplicate. The correlation coefficients and their probability levels were obtained from linear regression analyses. Differences between means were assessed using the Student’s t-test or analysis of variance with an α of 95%. For the BCT calculation (%), the ratio between the compound content in the mixed micelles and the initial extract corresponds to% BCT [32]. The content of each compound detected in the mixed micelles is also expressed as µg. The results from the uptake by Cacao-2 cells are expressed as ng/mg cell protein. Data related to the cellular uptake are also expressed as a percentage, i.e., as the ratio between the compound content absorbed by the cells and the inoculated content, which is related to the fraction of mixed micelles that were added to the cell culture medium.

## 3. Results

The present study adds to recent reports on the BCT and Caco-2 cellular uptake of carotenoids present in the fruit of *Bactris gasipaes* employing two models: (i) carotenoids obtained by COS extraction and (ii) carotenoids obtained with IL ([C_4_mim][BF_4_]). For alternative extracts, although 94% of the IL was recovered in the process developed (a very promising result considering the green chemistry approach), the missing 6%, which may be in the final extract, may impair the use of this extract as a food supplement. Thus, in this work, we investigated the concentration of the residual IL in the final extract, comparing the results of the extracts submitted to polishing with those where polishing was not performed (i.e., with a known concentration of IL present in the final extract). As depicted in Table 1, extracts that were submitted to the simulated digestion process did not have any IL concentration able to impair the applicability in the food sector (below the detection limit). On the contrary, in the extracts where IL was not withdrawn (control), approximately all IL used was quantified (98.37%), highlighting that the polishing process carried out in de Souza Mesquita et al. (2019) [10] was sufficiently effective for final application in the food sector.

As already reported in our most recent data studying peach palm carotenoids [10,31], the HPLC profile reported here was composed of all isomers of carotene and lycopene (Figure 2). The qualitative profile of micellarized carotenoids corresponds to the initial condition (emulsions before in vitro digestion), i.e., for both extracts, no carotenoid sub-products were detected at 450 nm after the in vitro approach (Figure 2). The BCT of the total carotenoids was better when carotenoids were extracted with IL. The micellization efficiency improved when using [C_4_min][BF_4_] for extraction (*t*-test: *p* < 0.05). More specifically, for the all-*trans* isomers, the micellization efficiency improved when the IL-mediated extraction was performed (Figure 3 and Table 1). Additionally, the microstructure (µm) of the initial emulsions made with both extracts were equivalent (*t*-test: *p* = 0.5896) (Table 2), suggesting that no initial physical differentiation impaired the comparison of the results. The same happened in all further phases, namely, oral, gastric, and intestinal (*t*-test: *p*_oral_ = 0.5763; *p*_gastric_ = 0.1519; *p*_intestinal_ = 0.3043). Figure 4 shows that the emulsions produced with the extracts (IL- and COS-mediated extracts) have a high polydispersity of the droplets, with sizes ranging between 7.84 and 18.87 µm. Besides, in Figure 4A–D, it is possible to note that no evident difference in the droplet dispersion was observed between the initial and oral phases for both extracts. However, the dispersion of the droplets changed according to the further stages of the digestion process. In the gastric stage, smaller droplets began to appear, due to the action of gastric enzymes, but some droplets of larger size were still present (Figure 4E,F). In the end, in the intestinal stage, only small droplets dispersed among themselves were found (~5 µm) (Table 2; Figure 4G,H).

Table 3 presents the carotenoid content data for both extracts at the initial, BCT, and Caco-2 uptake stages. The most abundant carotenoid in the initial condition, all-*trans*-β-carotene, was also the most micellarized (in terms of quantity). Figure 5 shows that the total content of carotenoids in the initial samples and the quantity of the carotenoids incorporated into micelles were correlated for both extracts ([C_4_mim][BF_4_]: R^2^ = 0.9945, r (Pearson) = 0.9972, *p* < 0.0001; Acetone: R^2^ = 0.9861, r (Pearson) = 0.9931, *p* < 0.0001). This correlation suggests that regardless of the type of carotenoid present in the peach palm fruits, there was a robust linear correlation between the initial and micellar carotenoid concentrations, which was also independent of the kind of extract, in this case, [C_4_mim][BF_4_] or acetone extracts. Figure 6 represents the cellular uptake (ng_carotenoids_/mg_cell protein_) of the carotenoids present in peach palm fruits. In terms of the total amount of carotenoids capitated by the Caco-2 cell monolayer, all-*trans*-β-carotene (all-*E*-βC) and all-*trans*-lycopene (all-*E-*L) were the most representative carotenoids accumulated by Caco-2 cells (Table 3). Overall, there was a significant difference (*t*-test: *p* < 0.05) in the cellular capitation of carotenoids extracted by [C_4_min][BF_4_] and acetone (Figure 6). The cellular uptake of carotenoids extracted with IL was 552.17 ng_carotenoids_/mg_cell protein_, which is 1.4-fold higher than the uptake of carotenoids extracted with acetone.

## 4. Discussion

The high extractive efficiency of ILs for obtaining many bioactive compounds has already been reported [6]; however, few studies to date have evaluated whether there are modifications of the biological effect of compounds extracted with these solvents [27]. Therefore, ILs represent a promising alternative for COS replacement in terms of environmental impacts and health risks and are excellent candidates for obtaining more eco-friendly value-added compounds [6]. In addition, several studies [10,11,12] have concluded that the thermal stability of all-*E*-βC, all-*E*-L, and xanthophyll carotenoids is better when extracted by ILs than by COS (obtained by acetone and ether mixtures), a very promising characteristic for the implementation of these compounds as natural food dyes. Furthermore, according to Bogacz-Radomska and Harasym (2018) [33], all-*E*-βC is produced on a large scale through chemical synthesis, but some studies have shown that this carotenoid obtained from natural sources has higher BCT and provides greater consumer confidence in the market than those of its synthetic counterpart. Thus, an alternative method using ILs to obtain carotenoids is highly desirable. As recently discussed, due to their non-corrosive characteristic and green and eco-friendly nature, ILs are an interesting substitute to the various conventionally used COSs in many fields, not only in extraction processes but also as catalysts for a wide range of organic transformation or synthetic procedures [34]. However, most of the studies have used erroneous judgment when considering ILs as non-toxic solvents, as most of them, including [C_4_mim][BF_4_], have some toxic potential (as highlighted on the label of the commercial product). For this, for example, if the target use of the obtained extract is the food sector, all IL must be removed (or the maximum possible that does not cause health side effects). In our previous work [10], we recovered 94% of the total IL used in the extraction process. However, in this work, we proved that the missing 6% IL is lost during the process, for example, during ultrasound homogenization, or retained in flasks, filters, or even in the biomass residue, and are not present in the final extract. This result opens doors considering the utilization of this extract as a possible food supplement, as well as done by us in a recent article (de Souza Mesquita et al., 2021) [35]. We supplemented male Wistar rats with carotenoids obtained with IL ([C_4_mim][BF_4_] and VOS, and compared the biological effect in terms of inflammatory markers, liver toxicity, and oxidative stress, concluding that the animals supplemented with carotenoids obtained with IL showed anti-inflammatory potential and antioxidant defense. The opposite happened with the animals supplemented with carotenoids obtained with VOS, which displayed liver toxicity (evidenced by liver steatosis) associated with oxidative stress damage (increase in levels of malondialdehyde and carbonyl protein). Thus, despite other alternative solvents having better credentials regarding human safety, imidazolium-based ILs are interesting for application when completely withdrawn from the extract, and even better when recycled in new extraction processes, which mitigates the environmental impact of the process (expressed by a lower carbon footprint compared with processes mediated by VOS), as done in de Souza Mesquita et al. (2019) [10]. Similarly, Martins et al. (2016) [36] evaluated the genotoxicity, mutagenicity, and cytotoxicity of carotenoids obtained with 1-Butyl-3-methylimidazolium chloride ([C_4_mim]Cl) in multiple organs of Wistar rats, and concluded that the carotenoids-extracted IL did not induce any toxicological effect. Additionally, as discussed by Murador et al. (2019) [27], it is more important to evaluate the toxicity of the product generated, in this case the carotenoids’ extracts, than to evaluate the toxicity of the solvent used in each process.

Our research group, also using a simulated digestion approach, reported that carotenoids released from *B. gasipaes* fruits (lyophilized samples) are lower than 10% [32]. Thus, in this study, to improve the BCT of these carotenoids, we applied pure extracts simulating food colorants in emulsions performed with the proper extract. Since the carotenoids were already extracted from the food matrix, we consider this an advantage for understanding the main phenomenon involved in this specific carotenoid release and cellular capitation. The extracts were emulsified (with no microstructure difference between emulsions performed using IL or acetonic extracts, as exposed in Figure 4 and Table 2), i.e., no physical barrier, such as fibers and proteins from the vegetal source was impeditive to assess the BCT. Additionally, some authors have reported different BCT values for all-*E*-βC, i.e., between 10% and 65% [33,37], which corroborates the results in this work (34.13 ± 4.30% for the [C_4_mim][BF_4_] extract and 27.31 ± 2.83% for the acetone extract). Our data demonstrate that when an IL was used to obtain carotenoids from *B. gasipaes*, there was a significant increase in the BCT and cellular uptake of carotenoids. We attributed these results to the capacity of the IL ([C_4_mim][BF_4_]) to recover other compounds (e.g., isomers of tocopherol) and other lipids present in *B. gasipaes* fruits [10], which could positively modulate carotenoid micellization. Another hypothesis was postulated based on the lack of selectivity in the acetone-mediated extraction. IL promotes highly selective extractions, i.e., no dissolution of significant amounts of carbohydrates or polysaccharides from the biomass [31,38]. Therefore, compounds that decrease micellization efficiencies, such as micronutrients, minerals, and carbohydrates, are not extracted, justifying the data obtained in this study [39]. The carotenoids present in *B. gasipaes* fruits are located in fat globules [21], which suggests the co-extraction of lipids, and other fat-soluble micronutrients, such as tocopherols isomers, present in *B. gasipaes* fruits [40]. On the other hand, the acetone-mediated process also extracts other compounds that impair the BCT of carotenoids, such as divalent ions, pectins, and fibers [39], which are also very common in *B. gasipaes* biomass. In our recent article, we corroborated the lack of efficiency of the COS-mediated process (acetone, ethanol, and ether mixtures) [31], which reinforces the prerogative exposed here.

Many studies have attempted to explain why carotenoids have low BCT and BV compared to those of other fat-soluble compounds. Considering this, it is possible to design new strategies to modulate these properties. One of the main hypotheses that corroborates the low absorption of carotenoids can be attributed to the physicochemical structure of the food matrix, mainly due to the rigid cell wall of the plants, as well as the shape of the chromoplast in which carotenoids are inserted into the cytoplasm and the presence of fibers [39]. Other authors have attributed the low BCT of these compounds to their liposolubility since they have a limited interface with aqueous micelles, making their intestinal absorption difficult [41]. According to Schweiggert et al. (2011) [42], depending on the genesis, carotenoid crystals may be stored in many kinds of structures, and this may alter the intestinal absorption of these pigments. Other authors reported that the carotenoids present in the fruit of *B. gasipaes* are allocated in fat globules and therefore are more bioavailable than carotenoids present in raphide crystals, such as all-*E*-L in tomatoes and all-*E*-βC in carrots [21]. The dissolution of carotenoids into lipids is necessary for their absorption, and carotenoids present in greasy food matrixes, such as emulsions, seem to be a great strategy for increasing their absorption. It was already reported that increasing the content of canola oil from 5% to 10% in carotenoid-fortified porridges resulted in a 5-fold higher micellization efficiency [43]. Additionally, some authors also described this phenomenon, concluding that BV of carotenoids is higher from salads ingested with full-fat than with fat-reduced salad dressings [44]. Thus, dietary lipids, mainly unsaturated fatty acids, such as those present in the sunflower oil used to produce the emulsions of the extracts used in this study (for IL and acetone extracts), increased the micellization and capitation efficiency of carotenoids [45].

As already stated, the food matrix plays a significant role in BCT and cellular Caco-2 uptake of carotenoids because it is the primary factor limiting adequate release of carotenoids being achieved [46]. Some studies have evaluated how processing and food preparation techniques influence carotenoid BCT, and the results mainly depend on the type of food matrix. Methods, such as prolonged/intense heating and pasteurization, can negatively influence carotenoid BCT since carotenoids are extremely sensitive to heat variation [47]. However, other authors demonstrated that depending on the type of carotenoid, some preparation techniques may increase the micellization efficiency, such as steaming, microwaving, and stir-frying [48]. Thus, many strategies for increasing the BCT and BV of these pigments have already been evaluated, but some gaps currently remain. Besides, the physicochemical properties of carotenoids, such as polarity, appear to be an essential factor in assessing BCT [49]; that is, hydrophobic pigments are less available for intestinal absorption. Some studies have reported that among the carotenoids, those that are hydroxylated (xanthophyll) have a higher rate of micellization compared to that of the all-isomers of carotene and lycopene [50,51]. Some authors studied the BCT of pure carotenoids and concluded that their micellar ratio was related to their hydrophobicity, which was not observed in this work since all-*E*-L had a superior micellization efficiency compared to that of the others [52]. In this work, a strong linear correlation between the initial concentration of carotenoids (before in vitro digestion) and mixed micelles (after in vitro digestion) was observed, which suggests that it is possible to estimate the amount of bioaccessible carotenoids when pure, i.e., without food matrix interference. Under the same perspective, some authors using all-*E*-βC from cassava suggested that the content of carotenoids in the samples could be a useful marker for assessing BCT without the need for in vitro digestion and animal studies [53]. Thus, this article also acts as an incentive to promote more experiments regarding the BCT and cellular uptake of compounds extracted using ILs, and other alternative solvents, such as eutectic solvents, aqueous solutions of surfactants, and more recently edible oils (used as promising solvents in the food sector) [15].

## 5. Conclusions

The intestinal assimilation model based on the Caco-2 cell line used in this study supports an understanding of the phenomenon of the cellular uptake of pure carotenoids obtained from COS and ILs, more specifically regarding all isomers of β-carotene and lycopene. The assays suggest a strong linear correlation between the initial concentration of all carotenoids present in peach palm fruits and the concentration of carotenoids in the mixed micellar phase for both extracts, which could assist further analysis using pure carotenoids. Additionally, we showed that adding carotenoids as food colorants may be an interesting strategy to increase their BCT. Thus, we proved that carotenoids extracted with ILs had higher BCT than acetone-extracted carotenoids, which was consequently reflected in a higher cellular uptake. To our knowledge, this is the first study that reports the results of BCT and cellular uptake of pure carotenoids from *B. gasipaes*, without interference from the food matrix, which assists in understanding the phenomena considering (only) the chemical structure of the carotenoid, in this case, all-isomers of carotene and lycopene. Furthermore, with this work, we hope that ILs are viewed as feasible alternatives as extraction solvents in the food industry and encourage further research to assess the possible biological effect and toxicity of the extracts obtained by them. Therefore, an integrative approach regarding a vast plethora of carotenoids is necessary to understand the phenomenon involved in BCT and BV, which could assist IL application in new fields.

## Figures and Tables

**Figure 1 molecules-26-03989-f001:**
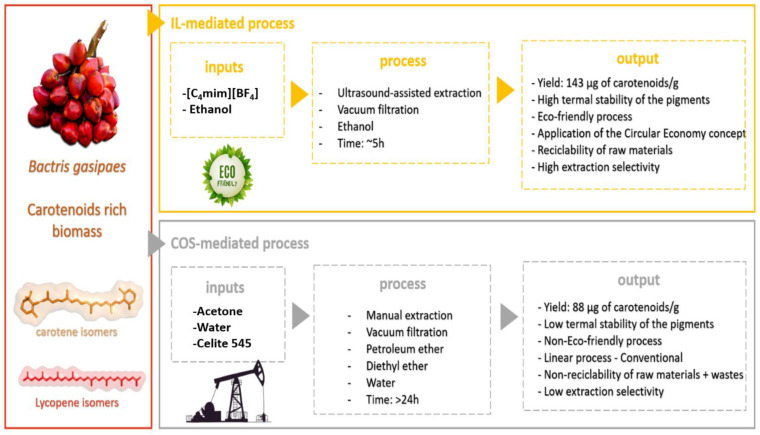
Schematic representation of the extraction processes evaluated in this study. The IL-mediated process is considered an eco-friendly method; and the COS-mediated process, based on organic solvents derived from petroleum.

**Figure 2 molecules-26-03989-f002:**
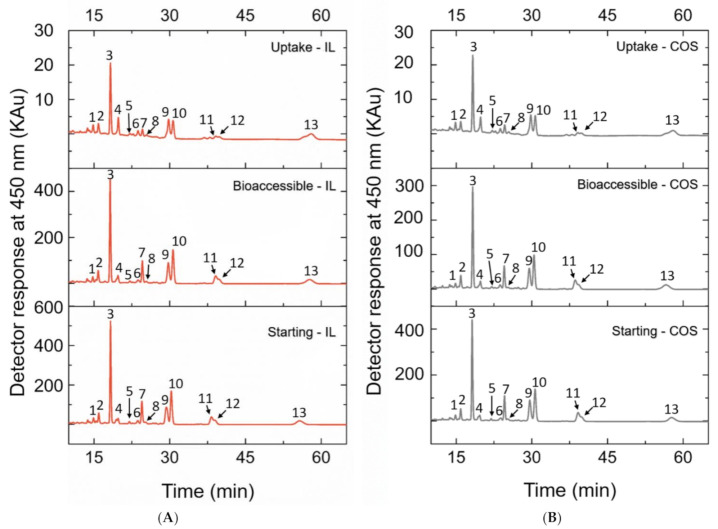
Chromatograms obtained by HPLC-PDA of carotenoid extracts from peach palm fruits via (**A**) IL xtraction; (**B**) COS extraction. For both extracts, the chromatograms are of the starting (before in vitro digestion), bioaccessible phase, and uptake samples. Peak 1: 13-*cis*-β-carotene; Peak2: all-*trans*-α-carotene; Peak 3: all-*trans*-β-carotene; Peak 4: *cis*-δ-carotene (1); Peak 5: *cis*-δ-carotene (2); Peak 6: *cis*-δ-carotene (3); Peak 7: all-*trans*-δ-carotene; Peak 8: *cis*-γ-carotene (1); Peak 9: all-*trans*-γ-carotene; Peak 10: *cis*-γ-carotene (2); Peak 11: *cis*-γ-carotene (3); Peak 12: 9-*cis*-lycopene; Peak 13: all-*trans*-lycopene.

**Figure 3 molecules-26-03989-f003:**
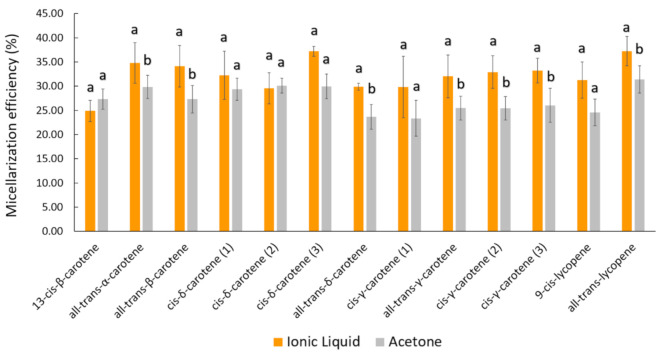
Bioaccessibility of carotenoids from peach palm fruits. Data represent the percentage of carotenoids incorporated into mixed micelles after in vitro digestion. Different letters above each carotenoid bar correspond to a significant difference between the [C_4_mim][BF_4_] and acetone extracts. *t*-test and Dunn test for means comparison were carried out.

**Figure 4 molecules-26-03989-f004:**
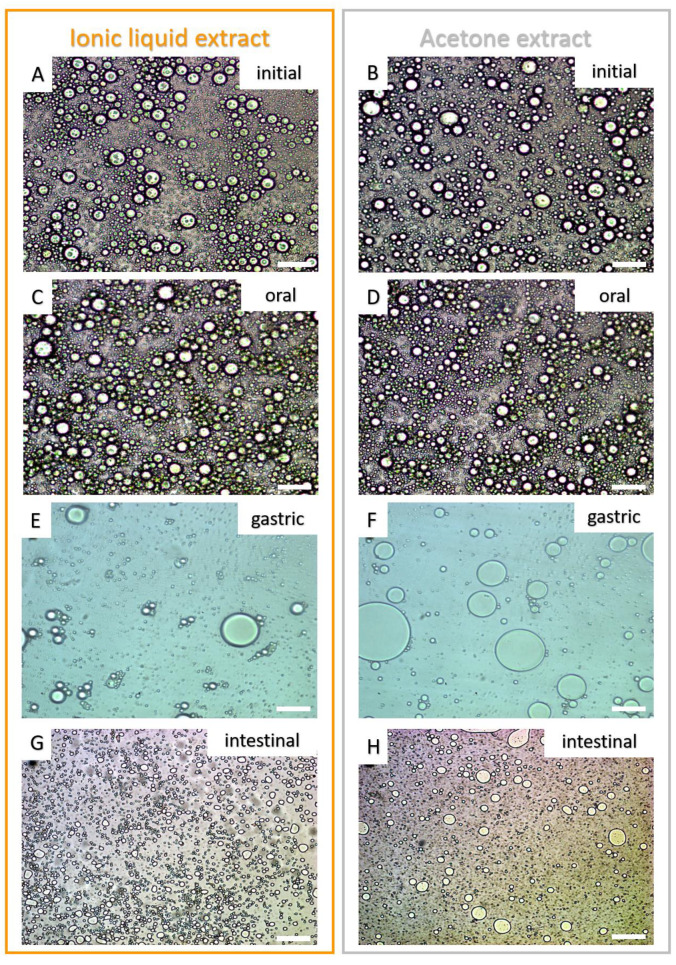
Micrographs of the initial emulsions (**A**,**B**) and after the simulated digestion process (oral—(**C**,**D**), gastric—(**E**,**F**), and intestinal phases—(**G**,**H**)). White scale bars represent 20 µm.

**Figure 5 molecules-26-03989-f005:**
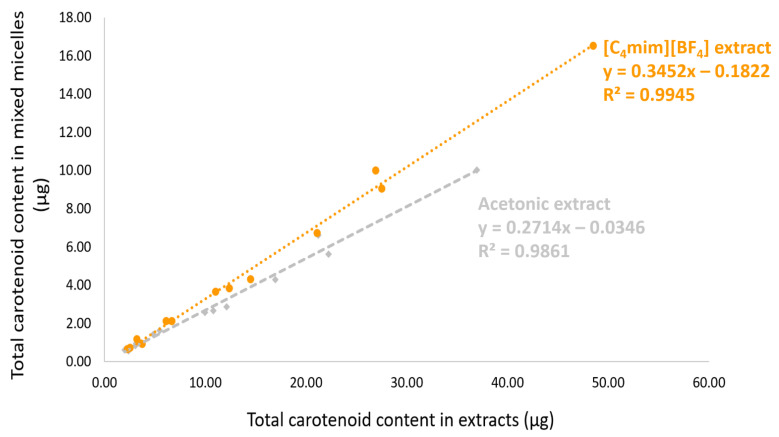
The total carotenoid content in the micelles of the digested samples correlated with the total carotenoid content of the initial samples. Data were analyzed by Pearson’s correlation. Orange points: [C_4_mim][BF_4_] extract; gray points: acetone extract.

**Figure 6 molecules-26-03989-f006:**
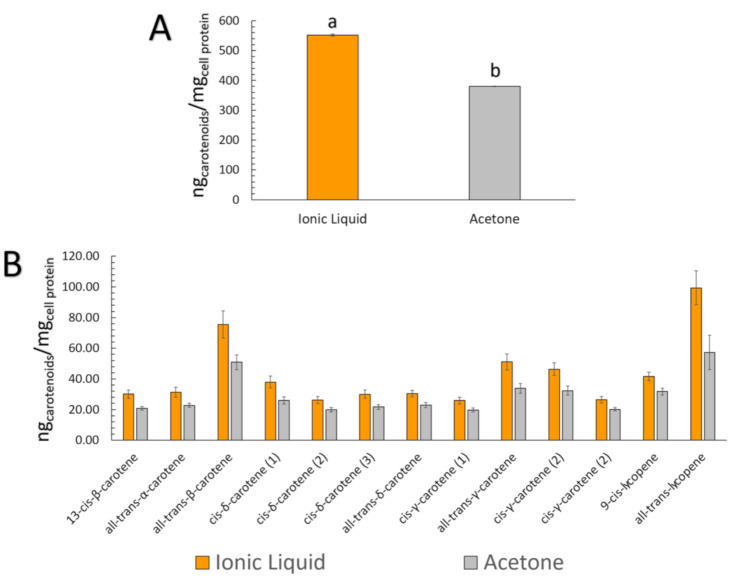
Carotenoid uptake by Caco-2 cells (ng_carotenoids_/mg_cell protein_). Data are related to the ng of carotenoids that were taken from the micellar fraction after 4 h of exposure. (**A**) Total carotenoids uptake from each extract—Different letters above each bar correspond to a significant difference between the [C_4_mim][BF_4_] and acetone extracts (*t*-test and Dunn test for mean comparisons were carried out); (**B**) Carotenoids uptake from each compound present in the extracts.

**Table 1 molecules-26-03989-t001:** Quantification of [C_4_mim][BF_4_] in carotenoid extract.

Samples	Initial Concentration—Input of IL	Concentration Recovered after Extraction Procedure (µg/mL)	Recovery (%)
Carotenoid extract submitted to polishing step (used in the simulated digestion process)	500 µg/mL	-	-
		-	
		-	
Carotenoid extract without polishing step (control)	500 µg/mL	491.22	98.37 ± 0.13
		492.35	
		492.14	

“-“ below detection limit.

**Table 2 molecules-26-03989-t002:** Microstructure values (µm) of the emulsions produced with both extracts (ionic liquid [C_4_mim][BF_4_], and acetone), and their respective values in each phase of the simulated digestion process (oral, gastric, and intestinal).

Samples	[C_4_mim][BF_4_] Carotenoid Extract (µm)	Acetone Carotenoid Extract (µm)
Initial (before in vitro digestion)	12.46 ± 2.96 a	11.98 ± 1.96 a
Oral	11.46 ± 1.63 a	11.13 ± 1.63 a
Gastric	10.8 ± 5.89 a	14.14 ± 6.55 a
Intestinal	5.01 ± 1.39 a	5.62 ± 1.77 a

The same letters in the lines represent statistical similarity (*t*-test: *p* > 0.05).

**Table 3 molecules-26-03989-t003:** Initial carotenoid content in the initial stage (emulsion samples), micelles fractions after in vitro digestion (BCT), and carotenoids absorbed (uptake).

Peak	Carotenoid	[C_4_mim][BF_4_] Carotenoid Extract			Acetone Carotenoid Extract		
		Starting (µg) *	Bioaccessible—BCT (µg) *^a^	Uptake (ng_carotenoids_/mg_cell protein_) *	Starting (µg) *	Bioaccessible—BCT (µg) *^a^	Uptake (ng_carotenoids_/mg_cell protein_) *
1	13-*cis*-β-carotene	3.71 ± 0.52	0.93 ± 0.21 (24.90%)	30.07 ± 2.60	3.07 ± 0.25	0.84 ± 0.05 (27.35%)	20.84 ± 1.22
2	all-*trans*-α-carotene	6.11 ± 0.17	2.12 ± 0.25 (34.80%)	31.32 ± 3.29	5.48 ± 0.66	1.63 ± 0.16 (29.85%)	22.78 ± 1.24
3	all-*trans*-β-carotene	48.50 ± 1.69	16.52 ± 1.80 (34.13%)	75.53 ± 8.92	36.98 ± 5.62	10.02 ± 1.20 (27.31%)	50.86 ± 4.88
4	*cis*-δ-carotene (1)	6.67 ± 0.52	2.13 ± 0.18 (32.24%)	37.97 ± 3.89	4.83 ± 0.39	1.41 ± 0.14 (29.33%)	26.07 ± 2.33
5	*cis*-δ-carotene (2)	2.51 ± 0.13	0.74 ± 0.04 (29.56%)	26.12 ± 2.31	2.00 ± 0.12	0.60 ± 0.03 (30.10%)	19.91 ± 1.34
6	*cis*-δ-carotene (3)	3.21 ± 0.07	1.19 ± 0.01 (37.21%)	29.99 ± 2.69	3.15 ± 0.31	0.94 ± 0.11 (29.96%)	21.70 ± 1.52
7	all-*trans*-δ-carotene	14.48 ± 0.74	4.33 ± 0.25 (29.88%)	30.44 ± 2.06	12.14 ± 1.70	2.86 ± 0.36 (23.67%)	22.94 ± 1.60
8	*cis*-γ-carotene (1)	2.26 ± 0.11	0.67 ± 0.11 (29.80%)	25.85 ± 2.12	2.39 ± 0.19	0.56 ± 0.08 (23.36%)	19.70 ± 1.30
9	all-*trans*-γ-carotene	21.07 ± 0.67	6.73 ± 0.80 (32.02%)	51.10 ± 5.27	16.95 ± 2.37	4.29 ± 0.46 (25.46%)	33.82 ± 3.12
10	*cis*-γ-carotene (2)	27.52 ± 0.98	9.06 ± 0.92 (32.92%)	46.40 ± 4.13	22.25 ± 3.34	5.62 ± 0.59 (25.42%)	32.32 ± 2.93
11	*cis*-γ-carotene (3)	11.03 ± 0.41	3.66 ± 0.31 (33.21%)	26.36 ± 2.08	9.98 ± 1.69	2.56 ± 0.14 (26.04%)	20.23 ± 1.11
12	9-*cis*-lycopene	12.35 ± 0.43	3.85 ± 0.42 (31.24%)	41.64 ± 2.84	10.81 ± 1.47	2.66 ± 0.45 (24.57%)	31.69 ± 2.32
13	all-*trans*-lycopene	26.90 ± 1.13	10.01 ± 0.86 (37.22%)	99.37 ± 11.15	21.23 ± 3.26	6.62 ± 0.76 (31.38%)	57.29 ± 11.13
Total		186.30 ± 5.39	61.95 ± 5.50 (33.25%) #	552.17 ± 3.65##	151.25 ± 21.15	40.59 ± 4.42 (26.84%)	380.16 ± 1.71

* Data are the mean ± standard deviation of three samples; ^a^ The percentage value is related to the initial mass of the emulsion; ^#^ *p* < 0.05 when compared with the BCT of the acetone carotenoid extract; ^##^ *p* < 0.05 when compared with uptake of the acetone carotenoid extract.

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
