# Peer review of "Bioaccessibility and Cellular Uptake of Carotenoids Extracted from Bactris gasipaes Fruit: Differences between Conventional and Ionic Liquid-Mediated Extraction"

_molecules, 2021, doi:10.3390/molecules26133989_

Round 1
Reviewer 1 Report
The main goal of the paper is optimization of extraction process but no new insight of the studied systems is provided. Essentially I have not learned much except from getting recipe of “how to”. I do not mean that authors did wrong job. To the contrary I appreciate their technical skills and undertaking such comprehensive set of measurements. The manuscript is publishable, but I strongly suggest selecting another journal than Molecules, which in my opinion, does not restricts itself to technical reports.
Author Response
Reviewer #1:
- “The main goal of the paper is optimization of extraction process but no new insight of the studied systems is provided. Essentially I have not learned much except from getting recipe of “how to”. I do not mean that authors did wrong job. To the contrary I appreciate their technical skills and undertaking such comprehensive set of measurements. The manuscript is publishable, but I strongly suggest selecting another journal than Molecules, which in my opinion, does not restricts itself to technical reports.”
Response: The authors acknowledge the reviewer suggestion. However, the main goal of this paper is not the optimization of the extraction process. The main goal of this article is to highlight that the bioaccessibility and cellular uptake of carotenoids obtained with IL are different to those obtained with organic solvents, as highlighted in the title of the manuscript (Bioaccessibility and cellular uptake of carotenoids extracted from Bactris gasipaes fruit: differences between conventional and ionic liquid-mediated extraction). We invite you to reread the actual version of the manuscript. We added some information about the polishing processes to facilitate the understanding of the study.
Reviewer 2 Report
It is an interesting article on the extraction of Carotenoids from Bactris gasipaes.
It is very important to indicate the economic aspects of these processes.
Some recommendations:
Line 107, Figure 1, use “Acetone” instead of “acetone”
Line 191, use “CaCl2” instead of “CaCl2.
Line 199, use “NaHCO3” instead of “NaHCO3”.
Line 213, use “cm2” instead of “cm2”.
Line 226, use “W” instead of “w”.
Line 228, use “N2” instead of “N2”.
Improve Figure 5.
For the references: "DOI numbers (Digital Object Identifier) are not mandatory but highly encouraged".
Author Response
- It is an interesting article on the extraction of Carotenoids from Bactris gasipaes.”
The authors acknowledge the reviewer comment
- It is very important to indicate the economic aspects of these processes.”
Response: Dear reviewer 2, we agree with you when point those economic aspects are important. However, for this manuscript, the main proposal was highlighting the bioaccesibility and cellular uptake of carotenoids obtained from different processes. Some recent articles have been proposed that IL-mediated processes for recovering pigments from natural sources have better economic impact when comparing with processes mediated by organic solvents, especially when the IL is recycled.
Some recommendations:
- Line 107, Figure 1, use “Acetone” instead of “acetone”
Response: The authors acknowledge the reviewer suggestion and have done the correction as requested.
- Line 191, use “CaCl2” instead of “CaCl2.”
Response: The authors acknowledge the reviewer suggestion and have done the correction as requested.
- Line 199, use “NaHCO3” instead of “NaHCO3”.”
Response: The authors acknowledge the reviewer suggestion and have done the correction as requested.
- Line 213, use “cm2” instead of “cm2”.”
Response: The authors acknowledge the reviewer suggestion and have done the correction as requested.
- Line 226, use “W” instead of “w”.
Response: The authors acknowledge the reviewer suggestion and have done the correction as requested.
- Line 228, use “N2” instead of “N2”.
Response: The authors acknowledge the reviewer suggestion and have done the correction as requested.
- Improve Figure 5.
Response: The authors acknowledge the reviewer suggestion and have done the correction as requested.
- For the references: "DOI numbers (Digital Object Identifier) are not mandatory but highly encouraged".
Response: The authors acknowledge the reviewer suggestion and have done the correction as requested.
Reviewer 3 Report
Comments
In this study, Ionic Liquids (IL) and Conventional Organic Solvents (COS) were used to extract carotenoids from Bactris gasipaes fruit. The bioaccessibility and Caco-2 cellular uptake of carotenoids extracted with IL and COS were compared. This article provides new methods and ideas. However, there are some problems in the manuscript that need further revision before publication.
Detailed questions and suggestions are given below.
- What is the “Amazonian Bactris gasipaes fruits” and what family and genus it belongs to? Please briefly introduce the information of raw materials.
- Line 86: “In addition, the non-sustainable exploitation of Bactris gasipaes fruits contributes to the loss of genetic biodiversity in the Amazon region and the under exploitation of its economic potential, a severe consequence considering the current global situation.” This sentence puzzled me. What is “Bactris gasipaes fruits”? and what dose “non-sustainable exploitation” mean?
- Line 94: Cost is also an extremely important factor in the processing process. Did the author compare the costs of the two extraction methods?
- Line 143: Please make a supplement to the experimental method on carotenoid polishing step. Author mentioned it several times in the article.
- Line 169: “Considering the solubility of the carotenoids in oily media, both extracts (mediated 168 by acetone and [C4mim][BF4]) were firstly resuspended in sunflower oil with the aid of 169 magnetic stirring (37 °C).” Why sunflower seed oil was selected to dissolve carotenoids, and whether the test effect would be affected.
- Line 246: “The results from the uptake by Cacao-2 cells are expressed as ng/mg cell protein.” Although overall well-written, the manuscript needs a recheck to tackle minor mistakes.
- Line 281: Please explain in the manuscript which extracts are corresponding to Figure 4 A-G.
- If the carotenoids extracted by IL method will be used in the food industry, It is necessary for the author to evaluate the safety of the extracted products.
- The novelty of this work should be highlight.
- The discussion section in the manuscript is quite sufficient, but some of the discussion is too redundant, please refine the language appropriately.
- There are many irregularities in the writing of the manuscript. Please recheck the manuscript carefully.
Author Response
In this study, Ionic Liquids (IL) and Conventional Organic Solvents (COS) were used to extract carotenoids from Bactris gasipaes fruit. The bioaccessibility and Caco-2 cellular uptake of carotenoids extracted with IL and COS were compared. This article provides new methods and ideas. However, there are some problems in the manuscript that need further revision before publication.
Detailed questions and suggestions are given below.
- What is the “Amazonian Bactris gasipaes fruits” and what family and genus it belongs to? Please briefly introduce the information of raw materials.
Response: The authors acknowledge the reviewer suggestion and have added the requested information in lines 85-88.
- Line 86: “In addition, the non-sustainable exploitation of Bactris gasipaes fruits contributes to the loss of genetic biodiversity in the Amazon region and the under exploitation of its economic potential, a severe consequence considering the current global situation.” This sentence puzzled me. What is “Bactris gasipaes fruits”? and what dose “non-sustainable exploitation” mean?
Response: Dear reviewer 3, we added some information about the Bactris gasipaes biomass that facilitated the understanding of the information highlighted.
- Line 94: Cost is also an extremely important factor in the processing process. Did the author compare the costs of the two extraction methods?
Response: Dear reviewer 3, we did not compare the costs of the two extraction methods. However, some articles already done it and concluded that processes mediated by alternative solvents have economic advantages compared to the processes performed using conventional protocols. This becomes even more prominent when the used IL is recycled, as well as done for produce the carotenoid extract used in this manuscript.
- Line 143: Please make a supplement to the experimental method on carotenoid polishing step. Author mentioned it several times in the article.
Response: Dear reviewer 3, we added a figure (FIGURE S1) in the supplementary material highlighting the polishing step performed.
- Line 169: “Considering the solubility of the carotenoids in oily media, both extracts (mediated 168 by acetone and [C4mim][BF4]) were firstly resuspended in sunflower oil with the aid of 169 magnetic stirring (37 °C).” Why sunflower seed oil was selected to dissolve carotenoids, and whether the test effect would be affected.
Response: The sunflower oil was selected since have a great potential to completely dissolve the carotenoid extract. An additional work is needed to confirm if different kinds of oils will interfere in the phenomena. However, we already published an article using sunflower oil for making an edible emulsion containing carotenoids obtained from Bactris gasipaes fruits, and for this we chose to perform the emulsion using sunflower oil. It is important to highlight that the both extracts (IL and COS) were submitted to the same process, using the same reagents, which does not put in check the comparation of the results.
- Line 246: “The results from the uptake by Cacao-2 cells are expressed as ng/mg cell protein.” Although overall well-written, the manuscript needs a recheck to tackle minor mistakes.
Response: The authors acknowledge the reviewer suggestion.
- Line 281: Please explain in the manuscript which extracts are corresponding to Figure 4 A-G.
Response: The authors acknowledge the reviewer suggestion and have added the requested information in lines 283-291.
- If the carotenoids extracted by IL method will be used in the food industry, It is necessary for the author to evaluate the safety of the extracted products.
Response: You have a point when say that the safety of the extracted needs to be highlighted. However, considering that this article has the main goal “evaluate the bioaccessiblity and cellular uptake of carotenoids obtained by different solvents and processes”, we judge that we already have enough novelty for one manuscript. For the best of our knowledge, our research group is the first in leading with this theme. For example, we published a similar article covering this issue (Food Chemistry https://doi.org/10.1016/j.foodchem.2020.127818), but with different IL-mediated process, using orange-peels as natural source, which is rich in chlorophylls and Xanthophylls. In this article, using the same prerogative already validated, we cover the influence of IL-mediated process in bioaccesibility and cellular uptake of carotenoids of the carotene class, namely beta-carotene and lycopene derivatives.
Recently, we submitted a manuscript regarding the supplementation of carotenoids obtained with this same IL in obese male Wistar rats, and concluded that the animals supplemented with this extract shown anti-inflammatory potential, and antioxidant defense, which not happened in animals supplemented with carotenoids obtained with acetone and ether. Additionally, the histological analysis from liver did not shown any toxicity signal in the animals supplemented with carotenoids obtained with IL. Thus, despite other alternative solvents have better credentials regarding your safety, imidazolium-based ILs are interesting for be applicable when completely withdrawn from the extract, and even better when recycled in new extraction processes, which mitigate the environmental impact of the process, as exposed in our article published on Green Chemistry journal.
- The novelty of this work should be highlight.
Response: We performed some modifications that highlighted the movelty of this manuscript.
- The discussion section in the manuscript is quite sufficient, but some of the discussion is too redundant, please refine the language appropriately.
Response: Dear reviewer 3, you have a point when say that some parts of the discussion section is quite redundant. We made some cuts to improve this. However, we justify that the discussion was designed considering the explanation of each result reported in order to justify the phenomena.
- There are many irregularities in the writing of the manuscript. Please recheck the manuscript carefully.
Response: The authors acknowledge the reviewer suggestion.
Round 2
Reviewer 1 Report
I am sorry but the answers change nothing in my opinion. I still see submitted paper as merely technical report, well done but of little scientific value.
It is very likely that utilization almost any ionic liquid will lead to solubility/extraction improvement with respect of majority of organic solvents.
I have serious problem with recommendation because there are not option to choose, which are adequate for this kind of study. My two concerns : bad selection of ionic liquid and lack of scientific insight still can be address to revised version.
I respect work but do not agree with presenting just technical reports in such prestigious journals as Molecules.
Author Response
Dear reviewer 1, as already said before, and also written in this manuscript, we know about the issues regarding the toxicity potential of imidazolium ILs. However, as explained in the text, we withdraw all IL from the carotenoid extract, as already quantified and explained in lines 257-269. Additionally, we added some information about the biological effect of this IL-based extract when used as a kind of supplement in rats in lines 269-279. These data are being evaluated by another prestigious journal and are under “minor reviews” now.
For your knowledge, we already tested other alternative solvents for recovery carotenoids from Bactris gasipaes. However, [C4mim][BF4] have the best extraction performance, with yields at least double of conventional organic solvents and even other alternative solvents, a very pertinent characteristic when a scaling-up is envisioned. Additionally, we performed the complete withdrawal of the used IL, i.e., no residual IL was present in the carotenoid extract (polishing step). Recently, we submitted a manuscript (in peer review status) regarding the supplementation of carotenoids obtained with this same IL in obese male Wistar rats and concluded that the animals supplemented with this extract shown anti-inflammatory potential, and antioxidant defense, which not happened in animals supplemented with carotenoids obtained with acetone and ether. Additionally, the histological analysis from the liver did not show any toxicity signal in the animals supplemented with carotenoids obtained with IL. Thus, despite other alternative solvents have better credentials regarding your safety, imidazolium-based ILs are interesting for be applicable when completely withdrawn from the extract, and even better when recycled in new extraction processes, which mitigate the environmental impact of the process, as exposed in our article published on Green Chemistry journal.